Journal of Data-centric Machine Learning Research (2024)     Submitted 11/23; Revised 03/24; Published 05/24

# DMLR: Data-centric Machine Learning Research
# -
# Past, Present and Future

**Luis Oala[1],** **Manil Maskey[2], Lilith Bat-Leah[3], Alicia Parrish[4], Nezihe Merve Gürel[5], Tzu-Sheng Kuo[6], Yang Liu[7,8], Rotem Dror[9], Danilo Brajovic[10], Xiaozhe Yao[34], Max Bartolo[11], William Gaviria Rojas[12], Ryan Hileman[13], Rainier Aliment[4], Michael W. Mahoney[14,15,16], Meg Risdal[17], Matthew Lease[18], Wojciech Samek[19,20], Debo Dutta[21], Curtis Northcutt[22], Cody Coleman[12], Braden Hancock[23], Bernard Koch[24], Girmaw Abebe Tadesse[25], Bojan Karlaš[26], Ahmed Alaa[14], Adji Bousso Dieng[27], Natasha Noy[4], Vijay Janapa Reddi[26], James Zou[28], Praveen Paritosh[29], Mihaela van der Schaar[30], Kurt Bollacker[29], Lora Aroyo[4], Ce Zhang[31,24], Joaquin Vanschoren[32], Isabelle Guyon[4,33,25], Peter Mattson[4,29]**

[1]*Dotphoton,* [2]*NASA,* [3]*Mod Op,* [4]*Google,* [5]*TU Delft,* [6]*Carnegie Mellon University,* [7]*UC Santa Cruz,* [8]*ByteDance Research,* [9]*University of Haifa,* [10]*Fraunhofer IPA,* [11]*Cohere,* [12]*CoactiveAI,* [13]*Talon,* [14]*UC Berkeley,* [15]*ICSI,* [16]*LBNL,* [17]*Kaggle,* [18]*UT Austin,* [19]*TU Berlin,* [20]*Fraunhofer HHI,* [21]*Nutanix,* [22]*Cleanlab,* [23]*Snorkel AI,* [24]*University of Chicago,* [25]*Microsoft AI for Good Lab,* [26]*Harvard University,* [27]*Princeton University,* [28]*Stanford University,* [29]*MLCommons,* [30]*University of Cambridge,* [31]*Together,* [32]*TU Eindhoven,* [33]*University of Paris-Saclay,* [34]*ETH Zurich,* [35]*ChaLearn*

**Reviewed on OpenReview:** *https://openreview.net/forum?id=2kpu78QdeE*

**Editor:** Hongyang Zhang

## Abstract

Drawing from discussions at the inaugural DMLR workshop at ICML 2023 and meetings prior, in this report we outline the relevance of community engagement and infrastructure development for the creation of next-generation public datasets that will advance machine learning science. We chart a path forward as a collective effort to sustain the creation and maintenance of these datasets and methods towards positive scientific, societal and business impact.

**Keywords:** data-centric machine learning, artificial intelligence, datasets, impact

## 1 Data Ambivalence in Machine Learning

Why state the obvious? Do we really need to emphasize some machine learning (ML) research as *data-centric*? Hasn't ML science, at its core, always been just that? After all, designing algorithms that extract models from data is machine learning's *summum bonum*. In the pursuit of this goal we often oscillate between two dominant phases: (i) design algorithm and throw data at it, (ii) go back to data (and its intermediate representations)

---

∗. To get involved in the community please join the discord at `https://discord.gg/FswYXMv4j9`. For updates to the manuscript you can contact `luis.oala@dotphoton.com`.

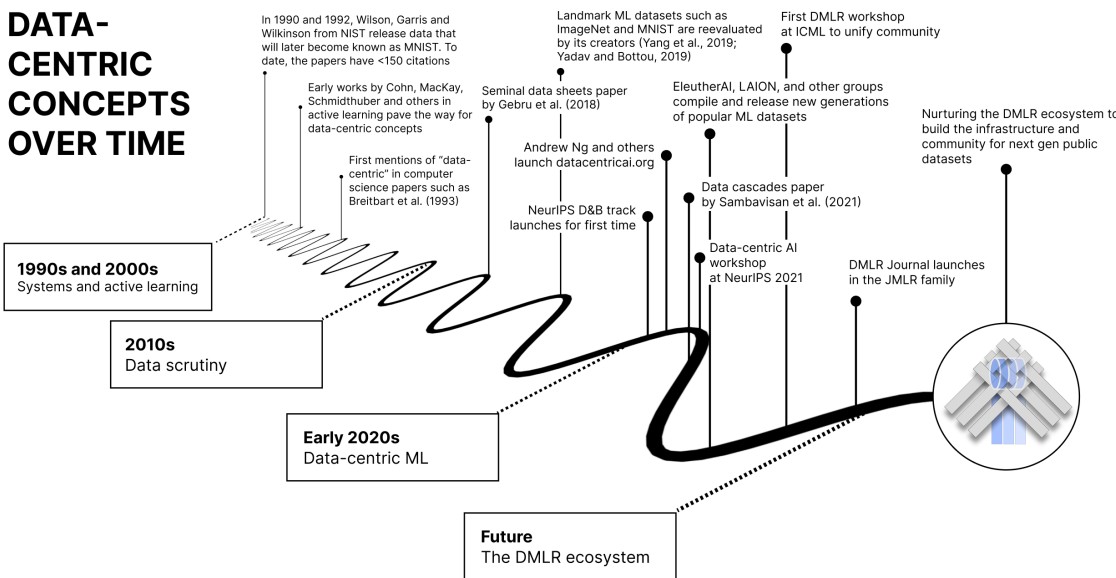

Figure 1: A timeline of some inflection points in the development of data-centric ideas.

to design better algorithm. This feedback loop informs the ambivalence towards data that many of us will encounter in machine learning practice: on the one hand, we want the algorithm to extract a model from data automatically; on the other hand, we often need to analyze the data and model manually to build good algorithms. Through the lens of this oscillation, data-centric machine learning research (DMLR) can broadly be described as infrastructure, methods and communities revolving around phase (ii).

In this editorial we outline key coordinates and objectives of DMLR, contextualize its origins, and summarize activities towards growing the DMLR ecosystem. And these lines are also an invitation, a call on you, the reader, to join us in shaping this DMLR future. Be it as open source contributor, community organizer, researcher or reviewer, your ideas and efforts are needed to maintain and shape DMLR further.

## 2 Past: Data-Centricity Over Time

Historically, the ambivalence towards data has manifested in different ways. In the early 1990s, Wilson, Garris and Wilkinson [1–4] distributed "*Handwriting Sampling Forms*" at the National Institute of Standards and Technology (NIST), digitizing the resulting data into the raw ingredients that were later turned into the now infamous machine learning staple MNIST. But as of October 4, 2023, their original publications have less than 150 citations combined. In comparison, the seminal LeNet paper by [5], which is often used as stand-in reference for the MNIST dataset, sits at 60,000 citations today.[1].

This is not to open artificial fault lines à la "*data people*" versus "*model people*". But one can wonder what such artifacts reveal about the incentives in machine learning and

---

1. As an aside, LeCun et al. [5] themselves did not cite the NIST prior works [1–4]. Notably, Yadav and Bottou [6] later revisited the history of MNIST.

how conducive they are to machine learning progress. Or whether, as [7] suggest, it slows progress because "*everyone wants to do the model work, not the data work*". Jumping to today, there are active and encouraging efforts in our community to counter this imbalance, prominently the *Datasets and Benchmark Track* at NeurIPS that was conceived for the first time in 2021 by Joaquin Vanschoren, Serena Yeung and Maria Xenochristou. We also have to emphasize that not all data work goes under-appreciated. One must only look at ImageNet [8] or CIFAR [9] for great success stories.

Algorithm development is often correlated with extended phases of "modality hegemony". Data staples for many early ML models were structured data, organized in tables, fueling the development of interpretable models that can handle discrete feature spaces effectively, such as the *Top-Down Induction of Decision Trees* (TDIDT) family of algorithms including CLS [10], ID3 [11], ACLS [12] or C4.5 [13]. In turn, leaps on less structured data, such as plain text or images, were accompanied by the development of new algorithms. This includes innovations on Convolutional Neural Networks (CNNs) for image classification such as AlexNet [14] or RNNs [15], LSTMs [16] and later transformer architectures [17] for text. Additionally, algorithms have been designed that can productively fuse different data modalities [18–21], port from one modality to another, such as transformers from text to vision tasks [22], or become fully modality agnostic by operating on byte representations [23]. Somewhat on the opposite side of the spectrum we also witness a resurgence of algorithm development for highly specific but widely adopted data modalities, such as structured tables [24–26] or data viewed as graphs [27; 28]. Critically, leaps in algorithm innovation typically presumed the existence of open datasets such as MNIST, ImageNet or CIFAR mentioned above. This is currently changing. For recent frontier algorithms, such as the OpenAI family of models [29; 30], the data acquisition and preparation is such a value-generating asset that it routinely remains closed off from public access. Exceptions do exist, especially by cooperative-style communities such as LAION [31], Common Crawl [32], or Eleuther [33], among others. To be clear, closed data assets are not new. However, in recent history they have increasingly driven frontier advances in machine learning systems which were typically powered by open datasets during the 2010s.

Around the same time as MNIST, the concept of data-centricity started to appear literally in early works by [34] and others. It was likely discussed in the systems and database circles long before the idea became an increasingly growing focus of research in the core machine learning community. The connection to systems persists to this day, evident in venues such as MLSys[2] or the DEEM workshops[3], due the high importance of optimized infrastructure to orchestrate and execute data transformation and machine learning workloads. Success stories can be found in frameworks to build and store models such as Torch [35], Theano [36], Caffe [37], TensorFlow [38], JAX [39] or PyTorch [40]. Sometimes these frameworks also led to optimized data formats, such as TensorFlow's TFRecord. Additionally, platforms like Kaggle, HuggingFace or OpenML have emerged as de-facto community data hubs and standardized data loading infrastructure. A new wave of emerging open-source projects such as Lance[4] aim to address existing gaps with respect to data loading needs. However, despite these advances, challenges regarding the compatibility,

---

2. https://mlsys.org/

3. http://deem-workshop.org/

4. https://github.com/lancedb/lance

mutability, and collaborability of datasets persist. Encouragingly, new initiatives, such as 'Croissant'[5][41], take stabs at the Babylonian tower of data formats, uniting key stakeholders in an effort to streamline data-centric machine learning infrastructure. Similarly, DataPerf [42] is a recent community-led benchmark suite for evaluating ML datasets and data-centric algorithms, enabling the ML community to iterate on datasets, instead of just architectures.

Alongside developments in infrastructure, we have over time also witnessed critical advances in the way datasets are collected, curated and maintained. From the beginnings of modern statistical science [43; 44], active learning, a set of methods concerned with data curation, has planted its roots firmly in the machine learning and statistical learning literature [45–51]. The next generation of machine learning datasets will further leverage these concepts, characterized by dense metadata annotation [52–54], collaborative refinement [55–57; 33], user preference and human feedback [58], and evolution over time [59; 60; 31], similar to the way we treat code for computer programs already today. Partially, this is already a lived reality in data catalogues like the Pile [33] or OpenWebMath [61]. Data provenance and ownership are also receiving increasing consideration by groups such as the "Data Trusts" initiative [62] and others. Our goal is to support the growth of the DMLR ecosystem into a strong community with effective infrastructure that will advance machine learning science through next generation datasets. These datasets shall serve as a bridge to connect fundamental problems (such as food insecurity and climate impact) with fundamental ML research by providing the right datasets for the right problems.

## 3 Present: Convening the Community at ICML 2023

In order to charge this effort, the data-centric ML community came together on July 29, 2023, in Honolulu, Hawaii, for the inaugural DMLR workshop at the International Conference on Machine Learning (ICML) 2023. The DMLR workshop was a point of convergence for previous activities including the Asilomar Datasets 2030 retreat, the Dataperf initiative[6], the NeurIPS 2021 data-centric AI workshop[7], the LAION community[8] and others. Invited speakers, panelists, poster presenters and attendees deliberated on the current state of data-centric machine learning and how we can advance the community and infrastructure towards the next generation of public machine learning datasets (see Figure 2 for a brief overview).

**Community engagement**    Andrew Ng concluded his keynote with open questions aimed at fostering further research and development in data-centric AI workflows. Isabelle Guyon proposed a peer-reviewed journal contributed to by AI-agents, aiming to foster scholarly community engagement. Dina Machuve discussed the role of community in data collection for agriculture in East Africa. Olga Rusnaskovsky and Vikram V. Ramaswamy addressed social bias in machine learning, calling for community action. The panel expressed substantial enthusiasm for the DMLR Journal, indicating a strong community interest in advancing the field. Paper authors highlighted diverse challenges in community standards ranging from

---

5. https://github.com/mlcommons/croissant

6. https://www.dataperf.org/

7. https://datacentricai.org/neurips21/

8. https://laion.ai/

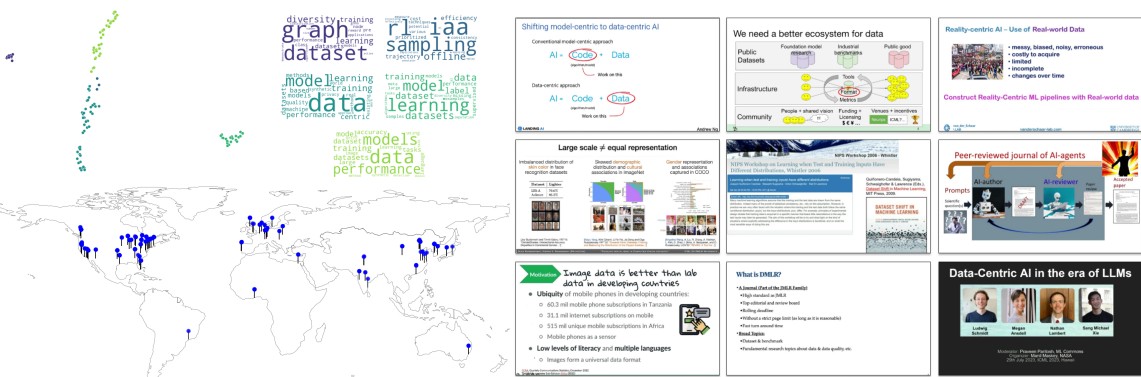

Figure 2: Themes and contributions from the community at the DMLR ICML 2023 workshop. **Top left:** LDA of accepted paper abstracts with `n_components` = 5. 2-d UMAP of LDA results which are 5-d corresponding to 5 components. Each dot represents an abstract, color coded by the most dominant topic identified by LDA. The topics identified by LDA are displayed alongside as top20 word clouds. **Bottom left:** A sample of the geographic coordinates of the institutions where authors of accepted works are based. It includes only those locations where the `geocode` API returns latitude and longitude information for fuzzy search on affiliation names (360 of 495 affiliations returned coordinates, note that not all 495 affiliations are unique). **Right:** Topics highlighted in the invited talk including prompt-based ML development (Andrew Ng), the DMLR ecosystem (Peter Mattson), reality-centric AI (Mihaela van der Schaar), bias in vision data (Olga Russakovsky and Vikram Ramaswamy), history of distribution shifts dating back to NeurIPS 2006 (Masashi Sugiyama), the AI research agent (Isabelle Guyon), nuances of data quality (Dina Machuve), the DMLR Journal (Ce Zhang) and data-centric LLMs (panel). Links to the full videos and slides of talks are available in Appendix C.

risk classification in driver telematics, the role of synthetic data in the scientific community, to the nuances of deep learning in neuroimaging and beyond.

**Infrastructure** Workshop contributions also illuminated the critical role of infrastructure in advancing data-centric machine learning. Andrew Ng emphasized the importance of rapid iteration cycles, facilitated by advancements in both theory and tools. Mihaela van der Schaar introduced tools like Data-IQ [63] for better data characterization. Peter Mattson and Praveen Paritosh discussed Croissant[9], a standardized dataset format, and DataPerf [42], an engine for refining datasets. Masashi Sugiyama added depth by discussing the complexities of machine learning models operating under distribution shifts. The panel, consisting of Ludwig Schmidt, Megan Ansdell, Nathan Lambert, and Sang Michael Xie, further emphasized that the development of systematic methods for constructing AI datasets is less advanced compared to model development but noted that tools and infrastructure are catching up. Poster presenters highlighted different aspects related to infrastructure such as quality control and streaming of distributed data.

---

9. `https://github.com/mlcommons/croissant`

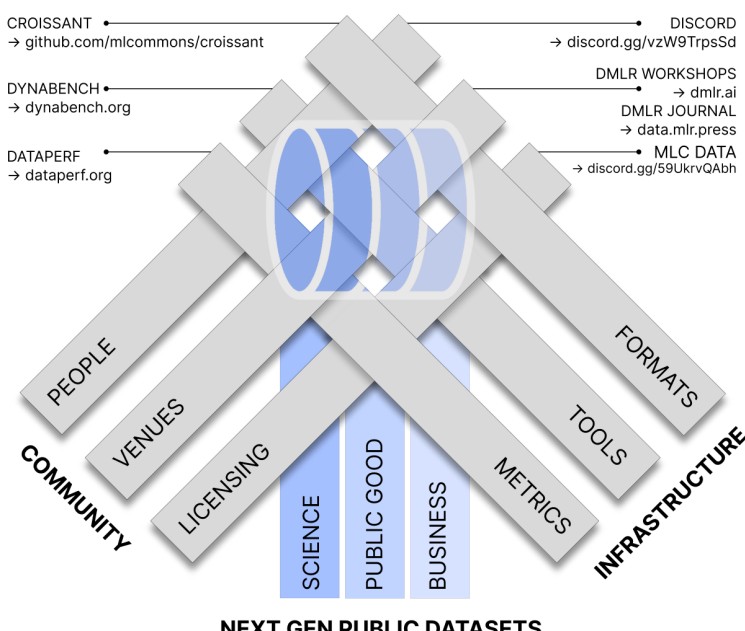

Figure 3: An overview of the DMLR ecosystem pillars and community projects.

**Datasets** The workshop participants also delved into the future of datasets in machine learning. Andrew Ng highlighted the growing relevance of small datasets and the practicality of few-shot learning techniques. Mihaela van der Schaar advocated for Reality-Centric AI [64]. Isabelle Guyon introduced AutoML+, a holistic system that includes data search, task definition, and preparation. Dina Machuve discussed the critical role of data in East African agriculture. The panel emphasized that data holds a central role in driving AI forward and highlighted the need for next-gen datasets to be more systematically constructed. Several papers also underscored the challenges and solutions in active learning, focusing on topics such as minimizing annotation cost and acquiring high-quality data for training discriminative models. Links to the full videos and slides of talks are available in Appendix C.

## 4 Future: Growing the DMLR Ecosystem

The field of machine learning is undergoing a profound transformation. While the past was characterized by the pursuit of innovative algorithms and architectures, the present and future pose growing data-centric questions. As large models become the norm and real-world efficacy becomes paramount, the emphasis is shifting towards the entire data lifecycle, from collection over storage and transformation to integration of results into other systems [65]. The importance of addressing societal issues through data further underscores this shift.

The role of the community in shaping the future of data-centric ML cannot be overstated. The recent DMLR workshop at ICML 2023 served as an inaugural meeting, igniting a spark for what is to come. A collective effort is required to create, enhance, and maintain public datasets. This involves establishing clear licensing protocols, technological standards, and fostering a culture of collaboration and shared, equitable ownership [66].

Earlier generations of machine learning datasets, such as MNIST, were often collected from scratch for specific pattern recognition tasks. Since then, crawling artifacts, for example ImageNet or the LAION datasets, have flourished and been scrutinized, introducing new questions on data provenance [67; 66], ownership, sharing and reviewing at scale [68]. These are not only philosophical questions but already slice of life, as "copyright haven" experiments such as in Japan [69] or litigation against commercial users of web crawled data [70] illustrate. Moving forward, alternative models for data ownership may warrant reconsideration. For example, data trusts [62] offer legal and operational frameworks to manage and govern access to data transparently. Testbeds for this practice can be found in places like Delhi's open traffic data [71], the European Union data sharing spaces [72] or the Swiss health data sharing platform SHDS [73]. In the context of machine learning, data trusts offer a structured approach to address issues of data privacy, security, and ownership, enabling collaborative and responsible data sharing among multiple stakeholders. By establishing clear rules and protocols for data usage, data trusts can incentivize the creation of new datasets while safeguarding sensitive information and intellectual property [74]. Such principles are not exclusively explored in policy, they also underpin technological experiments for transaction-driven, decentralized machine learning [75]. Interested contributors can already find several entry points, including

- DMLR workshops or tracks at the main machine learning conferences such as NeurIPS Datasets and Benchmarks
  https://dmlr.ai/

- DMLR journal as an author, editor, reviewer
  https://data.mlr.press/

- Data provenance and governance initiatives such as
  https://datatrusts.uk/ [62]
  https://dataprovenance.org/ [66]

- Socials and informal research retreats such as Asilomar Datasets 2030

- Open source data-centric libraries such as
  https://github.com/vanderschaarlab/datagnosis

Community building also needs to permeate the technical domains in which data-centric methods can be applied to realize real-world utility. Past advances in healthcare, finance, agriculture, climate science or recommender systems are testament to their potential of delivering real-world impact. These include federated learning frameworks, such as [76], enabling collaborative data engineering across institutions and enhancing predictive models while safe-guarding patients' and IP holders' data rights. In the financial domain, careful data curation has been pivotal in creating more robust and adaptive fraud detection systems. A notable example is the work by Dal Pozzolo et al. [77] which leverages vast, quality-curated transaction datasets to identify fraudulent activities with high accuracy. Precision

agriculture has benefited from crowd-sourced, quality controlled datasets, too. This spans satellite imagery and sensor data fusion to optimize crop yield predictions or plant disease detection from images, serving as templates for the potential of community-sourced data in improving agricultural outcomes [78; 79]. In climate science models have been enhanced through careful data synthesis to provide more accurate predictions of weather patterns and climate change impacts, for example extreme weather events from large-scale climate simulations [80]. In recommender systems, the Netflix prize competition is an early example for how community engagement and collaborative filtering techniques can improve the accuracy of production systems [81]. Continued engagement of the application domains will be crucial to convert innovations from the DMLR community to real-world impact.

Furthermore, an infrastructure that supports the collaborative creation and enhancement of datasets is crucial. This infrastructure should champion the principles of open-source software, fostering a culture of shared responsibility and continuous improvement. The concept of "living datasets" emerges, emphasizing the dynamic nature of data [82; 83] and the importance of metrics [84–86] and rich, flexible metadata in ensuring its relevance. Exemplar activities that continuously onboard input from contributors include, among others

- Croissant dataset format
  `https://github.com/mlcommons/croissant` [41]

- Dynabench dynamic data collection and benchmarking platform
  `https://dynabench.org/`

- Dataperf, metrics for data-centric algorithm benchmarks
  `https://www.dataperf.org/home` [42]

Vibrant communities and innovative infrastructure will facilitate the future of machine learning datasets that cater to large models and real-world efficacy. These datasets should encapsulate the entire data lifecycle, ensuring they remain relevant and adaptable. They must be amenable enough to support the evolving research questions in machine learning. Furthermore, they should help address societal issues and allow analyses with respect representation and biases [65; 87]. The integrity of data forms the bedrock of reliable machine learning models. This involves addressing challenges related to noisy measurements, noisy labels and uncertainty [67; 88]. Ensuring the quality of data used for ML training and evaluation is paramount, as it directly influences the efficacy and reliability of the resulting models. New datasets, also called data++ [89] by some, thus should increasingly support the optimization of data itself [90–95] as part of the machine learning lifecycle. Ongoing initiatives that amalgamate these ingredients comprise, among others

- Machine Learning Common's datasets working group
  `https://mlcommons.org/en/groups/datasets/`

- UN's AI for Good SDG gateway in collaboration with DMLR
  `https://aiforgood.itu.int/about-ai-for-good/discovery/#Datacentric`

- Independent research collectives such as LAION or EleutherAI
  `https://laion.ai/`, `https://www.eleuther.ai/`

- A diversity of open source benchmarking and evaluation repositories such as
  `https://github.com/erichson/SuperBench`,

`https://wilds.stanford.edu/` [96],
`https://github.com/hendrycks/robustness`[97],
`https://github.com/basveeling/pcam` [98],
`https://mlcommons.org/en/dollar-street/` [99],
`https://github.com/modestyachts/ImageNetV2` [100],
`https://github.com/inverse-scaling/prize` [101]

Investing in public datasets offers a plethora of benefits. It has the potential to accelerate innovation in the field of ML, reduce legal and ethical risks associated with data usage, and address pressing societal challenges. The emphasis is on creating datasets that not only advance the field of ML but also contribute positively to society at large by addressing real-world problems. The DMLR community is already expansive and, even more importantly, ongoing. We envision an ecosystem that strengthens these pillars and supports the growth and funding of new ideas. Whether you are a researcher, a practitioner, or an enthusiast, your insights and contributions to DMLR are the determinants of the data-centric machine learning future.

## Continual learning without forgetting

With this editorial we aim to highlight critical developments in data-centric machine learning and provide an overview of entry points for contributions to different activities in the extended community. In a dynamic system, a snapshot like this editorial will always contain some approximation error. If you know of relevant resources that were omitted please do not be shy and reach out. We will be happy to update them.

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

## Appendix A. The people behind the DMLR program at ICML 2023

Next to the organizers, speakers and attendees, the DMLR community is made up of its reviewers and submitting authors. For the first DMLR meeting at ICML 2023 (https://dmlr.ai/) we are grateful to the following people for volunteering their time and expertise.

### A.1 Program committee

Ziniu Li (The Chinese University of Hong Kong, Shenzhen), Zhixin Huang (University of Kassel), Zhaowei Zhu (University of California, Santa Cruz), Yue Yu (Georgia Institute of Technology), Yue Xing (Michigan State University), Yuanshun Yao (ByteDance), Yoav Wald (Johns Hopkins), Yixin Liu (Monash University), Yilin Zhang (Meta), Yi-Fan Zhang (NLPR, China), Yang Liu (UC Santa Cruz), Xinhui Li (Georgia Institute of Technology), Xianling Zhang (Ford Motor Company), William Gaviria Rojas (Coactive AI), Usama Muneeb (University of Illinois Chicago), Tzu-Sheng Kuo (Carnegie Mellon University), Tom Viering (Delft University of Technology, Netherlands), Thao Nguyen (University of Washington), Sumedh Datar (UTA), Sigrid Passano Hellan (University of Edinburgh), Siddharth Joshi (UCLA), Si Chen (Virginia Tech), Shin'ya Yamaguchi (NTT / Kyoto University), Sebastian Schelter (University of Amsterdam), Roger Waleffe (University of Wisconsin-Madison), Rasool Fakoor (AWS), Puja Trivedi (University of Michigan), Praveen Paritosh (Google), Peter Mattson (Google), Paolo Climaco (Universitat Bonn), Oliver Lenz (Universiteit Gent), Nauman Ahad (Georgia Institute of Technology), Muhammed Razzak (University of Oxford), Min Du (Palo Alto Networks), Megan Richards (Meta), Mayee Chen (Stanford University), Manil Maskey (NASA MSFC), Madelon Hulsebos (University of Amsterdam), Luis Oala (Dotphoton AG), Linxin Song (Waseda University), Linus Ericsson (University of Edinburgh), Lilith Bat-Leah (N/A), Liangchen Luo (Google), Li Jiang (Tsinghua University), Lenora Gray (Redgrave Data), Kurt Bollacker (The Long Now Foundation), Karthick Gunasekaran (Researcher), Julian Bitterwolf (University of Tubingen), Jinyi Liu (Tianjin University), Jieyu Zhang (University of Washington), Jialu Wang (University of California, Santa Cruz), Jiaheng Wei (UCSC), Jiachen Wang (Princeton University), Jeyeon Eo (Soongsil University), Jerone Andrews (Sony AI), Jayaraman J. Thiagarajan (Lawrence Livermore National Laboratory), Jarne Van den Herrewegen (Oqton / Ghent University), Jan Van Rijn (Leiden University), Ian Beaver (Verint Systems Inc), Huaizheng Zhang (BreezeML), Himchan Jeong (Simon Fraser University), Hidetomo Sakaino (Weathernews Inc.), Harit Vishwakarma (University of Wisconsin Madison), Hao Cheng (University of California, Santa Cruz), Hang Zhou (UC Davis), Guozheng Ma (Tsinghua University), Gregory Yauney (Cornell University), Feiyang Kang (Virginia Tech), Fangyi Chen (Carnegie Mellon University), Dionysis Manousakas (Amazon), Diego Botache (University of Kassel), Danilo Brajovic (Fraunhofer), Daniel Galvez (NVIDIA), Chanjun Park (Upstage), Beverly Quon (University of California, Irvine), Andre Carreiro (Fraunhofer Portugal AICOS), Amro Abbas (Meta), Ali Hakimi Parizi (Thomson Reuters), Alexander Li (Carnegie Mellon University)

## A.2 Authors

Hashan Peiris (Simon Fraser University), Himchan Jeong (Simon Fraser University), Jae Kwang Kim (Iowa State University), Gantavya Bhatt (University of Washington, Seattle), Arnav Das (University of Washington), Megh M Bhalerao (University of Washington), Rui Yang (Memorial Sloan Kettering Cancer Center), Vianne R Gao (Weill Medical College), Jeff Bilmes (UW), Linxin Song (Waseda University), Jieyu Zhang (University of Washington), Xiaotian Lu (Kyoto University), Tianyi Zhou (University of Maryland, College Park), Yue Xing (Michigan State University), Ashutosh Pandey (Meta Platforms), David Yan (Meta Platforms), Fei Wu (Meta), Michael Fronda (Meta Platforms), Pamela Bhattacharya (Meta Platforms), Yongchao Zhou (University of Toronto), Hshmat U Sahak (University of Toronto), Jimmy Ba (University of Toronto), Eujeong Choi (Upstage), Chanjun Park (Upstage), NamHyeok Kim (Upstage), Damrin Kim (Konkuk University), Harksoo Kim (Konkuk University), Sang Michael Xie (Stanford University), Hieu Pham (Google), Xuanyi Dong (University of Technology Sydney), Nan Du (Google Brain), Hanxiao Liu (Google Brain), Yifeng Lu (Google Brain), Percy Liang (Stanford University), Quoc Le (Google Brain), Tengyu Ma (Stanford), Adams Wei Yu (Google Brain), Bohan Wang (University of Science and Technology of China), zhengyu hu (NA), Pang We Koh (University of Washington), Alexander J Ratner (University of Washington), Jifan Zhang (University of Wisconsin), Shuai Shao (Meta), Saurabh Verma (Meta), Robert Nowak (University of Wisconsin, Madison), Ziniu Li (The Chinese University of Hong Kong, Shenzhen), Tian Xu (Nanjing University), Zeyu Qin (HKUST), Yang Yu (Nanjing University), Zhiquan Luo (The Chinese University of Hong Kong, Shenzhen and Shenzhen Research Institute of Big Data), Seonmin Koo (Korea University), Seolhwa Lee (University of Copenhagen), Jaehyung Seo (Korea University), Sugyeong Eo (Korea University), Hyeonseok Moon (Korea University), Heuiseok Lim (Korea University), Piotr Przemielewski (Jagiellonian University), Witold Wydmański (Jagiellonian University), Marek Śmieja (Jagiellonian University), Hang Zhou (UC Davis), Jonas Mueller (Cleanlab), Mayank Kumar (Cleanlab), Jane-Ling Wang (UC Davis), Jing Lei (Carnegie Mellon University), Saad A Almohaimeed (University of Central Florida), Saleh Almohaimeed (University of Central Florida), Ashfaq Ali Shafin (Florida International University), Bogdan Carbunar (Florida International University), Ladislau Boloni (University of Central Florida), Yuanshun Yao (ByteDance), Yang Liu (UC Santa Cruz), Harit Vishwakarma (University of Wisconsin Madison), Heguang Lin (University of Wisconsin-Madison), Frederic Sala (University of Wisconsin-Madison), Ramya Korlakai Vinayak (University of Wisconsin-Madison), Jesse E Cummings (MIT), Elías Snorrason (Cleanlab), Seungjun Lee (Korea University), Stefan Grafberger (University of Amsterdam), Bojan Karlaš (Harvard University), Paul Groth (University of Amsterdam), Sebastian Schelter (University of Amsterdam), Huaizheng Zhang (BreezeML), Yizheng Huang (BreezeML), Yuanming Li (Independent Researcher), Jaeseung Heo (POSTECH), Seungbeom Lee (POSTECH), Sungsoo Ahn (POSTECH), Dongwoo Kim (POSTECH), Patrik Okanovic (ETH Zurich), Roger Waleffe (University of Wisconsin-Madison), Vasilis Mageirakos (ETH Zurich), Konstantinos Nikolakakis (Yale University), Amin Karbasi (Yale), Dionysios Kalogerias (Yale University), Nezihe Merve Gürel (ETH Zürich), Theodoros Rekatsinas (ETH Zurich), Si Chen (Virginia Tech), Feiyang Kang (Virginia Tech), Nikhil Abhyankar (Virginia Tech), Ming Jin

(Virginia Tech), Ruoxi Jia (Virginia Tech), Joshua L Vendrow (MIT), Saachi Jain (MIT), Logan Engstrom (MIT), Aleksander Madry (MIT), Hoang Anh Just (Virginia Tech), Anit Kumar Sahu (Amazon Alexa AI), Jinsung Kim (Korea University), Min Du (Palo Alto Networks), Nesime Tatbul (Intel Labs and MIT), Brian Rivers (Intel), Akhilesh Kumar Gupta (University of Pennsylvania), Lucas Hu (Palo Alto Networks), Wei Wang (Palo Alto Networks), Ryan C Marcus (MIT), Shengtian Zhou (Snap), Insup Lee (University of Pennsylvania), Justin Gottschlich (Merly and Stanford University), Paolo Climaco (Institut für Numerische Simulation, Universität Bonn), Jochen Garcke (University Bonn), Nathan Vaska (MIT Lincoln Laboratories), Victoria Helus (MIT Lincoln Laboratory), Natalie Abreu (MIT Lincoln Laboratory), Dahyun Jung (Korea University), Jaewook Lee (Korea University), Yue Yu (Georgia Institute of Technology), Yuchen Zhuang (Georgia Institute of Technology), Yu Meng (University of Illinois Urbana-Champaign), Ranjay Krishna (University of Washington), Jiaming Shen (Google Research), Chao Zhang (Georgia Institute of Technology), Jerone T A Andrews (Sony AI), Dora Zhao (Sony AI), William Thong (Sony AI), Apostolos Modas (Sony), Orestis Papakyriakopoulos (Sony AI), Alice Xiang (Sony AI), Lei Shu (Google), Liangchen Luo (Google), Jayakumar Hoskere (Google), Yun Zhu (Google), Yinxiao Liu (Google), Simon Tong (Google), Jindong Chen (Google), Lei Meng (Google), Yongchan Kwon (Columbia University), James Zou (Stanford University), Shivangana Rawat (Indian Institute of Technology, Hyderabad), Chaitanya Devaguptapu (Fujitsu Research), Vineeth Balasubramanian (Indian Institute of Technology Hyderabad), Jayaraman J. Thiagarajan (Lawrence Livermore National Laboratory), Vivek Narayanaswamy (Lawrence Livermore National Laboratory), Puja Trivedi (University of Michigan), Rushil Anirudh (Lawrence Livermore National Laboratory), Shreyas Krishnaswamy (University of California, Berkeley), Lisa Dunlap (UC Berkeley), Lingjiao Chen (University of Wisconsin-Madison), Matei Zaharia (Stanford and Databricks), Joey Gonzalez (Berkeley), Hao Cheng (University of California, Santa Cruz), Qingsong Wen (Alibaba DAMO Academy), Liang Sun (Alibaba Group), Mayee Chen (Stanford University), Nicholas Roberts (University of Wisconsin-Madison), Kush Bhatia (Stanford University), Jue WANG (Zhejiang University), Ce Zhang (ETH), Christopher Re (Stanford University), Andre V Carreiro (Fraunhofer Portugal AICOS), Mariana Pinto (Faculty of Science and Technology, Nova University of Lisbon), Pedro S Madeira (Fraunhofer Portugal AICOS), Alberto Lopez (Imprensa Nacional - Casa da Moeda), Hugo Gamboa (LIBPhys, Faculdade de Ciências e Tecnologia, Universidade Noval de Lisboa), M. Eren Akbiyik (ETH Zurich), Florian Grötschla (ETH Zürich), Beni Egressy (ETH Zurich), Roger Wattenhofer (ETH Zurich), Oliver U Lenz (Universiteit Gent), Daniel Peralta (Ghent University ), Chris Cornelis (Ghent University), Aabha Pingle (Pune Institute of Computer Technology), Aditya Vyawahare (Pune Institute of Computer Technology), Isha Joshi (Pune Institute of Computer Technology), Rahul Tangsali (SCTR's Pune Institute of Computer Technology), Raviraj Joshi (Indian Institute of Technology Madras), Jarne Van den Herrewegen (Oqton / Ghent University), Tom Tourwé (Oqton), Francis Wyffels (Ghent University), Julian Bitterwolf (University of Tübingen), Maximilian Mueller (University of Tübingen), Matthias Hein (University of Tübingen), Darlington Akogo (minoHealth), Issah A Samori (minoHealth AI Labs), Cyril S K Akafia (minoHealth AI Labs), Harriet Dede Fiagbor (minoHealth AI Labs), Andrews A Kangah (KaraAgro AI Labs), Donald Donald (KaraAgro), Kwabena Fuachie (Kara Agro AI), Luis Oala ( Dotphoton AG), Li Jiang (Tsinghua University), Sijie Chen (Fudan Uni-

versity), Jielin Qiu (Carnegie Mellon University), Haoran Xu (JD Technology), Victor Chan (TBSI), DING ZHAO (Carnegie Mellon University), Sigrid Passano Hellan (University of Edinburgh), Christopher Lucas (University of Edinburgh), Nigel Goddard (University of Edinburgh), Dionysis Manousakas (Amazon), Sergul Aydore (Amazon), Nauman Ahad (Georgia Institute of Technology), Namrata Nadagouda (Georgia Institute of Technology), Eva L Dyer (Georgia Tech), Mark Davenport (Georgia Institute of Technology), Rafael Mosquera Gómez (MLCommons), Julian Eusse (MLCommons), Juan Manual Ciro (Factored), Daniel Galvez (NVIDIA), Ryan Hileman (Talon Voice), Kurt Bollacker (The Long Now Foundation), David Kanter (MLCommons), Siddharth Joshi (UCLA), Baharan Mirzasoleiman (UCLA), Jeffrey Li (University of Washington), Ludwig Schmidt (University of Washington), Vedang Lad (MIT), Ammar Sherif (Nile University), Abubakar Abid (Hugging Face), Mustafa Elattar (Nile University), Mohamed ElHelw (Nile University), Ulyana Tkachenko (Cleanlab), Aditya Thyagarajan (CleanLab), Alycia Y Lee (Stanford University), Brando Miranda (Stanford University), Sanmi Koyejo (Stanford University), Xinhui Li (Georgia Institute of Technology), Alex Fedorov (Georgia Institute of Technology), Mrinal Mathur (Georgia State University), Anees Abrol (TReNDS), Gregory Kiar (Child Mind Institute), Sergey Plis (Georgia State University), Vince Calhoun (TReNDS), Patrick Yu (University of Illinois Urbana-Champaign), Saumya Goyal (Stanford University), Yu-Xiong Wang (University of Illinois at Urbana-Champaign), Beverly A Quon (University of California, Irvine), Jean-Luc Gaudiot (University of California, Mark Heimann (Lawrence Livermore), Danai Koutra (U Michigan), Yifang Chen (University of Washington), Gregory H Canal (University of Wisconsin-Madison), Stephen O Mussmann (University of Washington), Yinglun Zhu (University of Wisconsin-Madison), Simon Du (University of Washington), Kevin Jamieson (U Washington), Alexander C Li (Carnegie Mellon University), Ellis L Brown (Carnegie Mellon University), Alexei A Efros (UC Berkeley), Deepak Pathak (Carnegie Mellon University), Dingshuo Chen (University of Chinese Academy of Sciences), Yanqiao ZHU (University of California, Los Angeles), Yuanqi Du (Cornell University), Zhixun Li (The Chinese University of Hong Kong), Qiang Liu (Institute of Automation, Chinese Academy of Sciences), Shu Wu (NLPR, China), Liang Wang (NLPR, Joel Niklaus (University of Bern), Veton Matoshi (Bern University of Applied Sciences), Matthias Stürmer (University of Bern), Ilias Chalkidis (University of Copenhagen), Daniel Ho (Stanford Law), Siddarth Ramesh (Adobe), Surgan Jandial (MDSR Labs, Adobe), Gauri Gupta (MIT), Piyush Gupta (Adobe Systems India Pvt Ltd), Balaji Krishnamurthy (), Kushal Tirumala (FAIR), Daniel Simig (Meta AI), Armen Aghajanyan (FAIR), Ari S Morcos (Facebook AI Research (FAIR)), Yonghyun Kwon (Iowa State University), Rohith Peddi (The University of Texas at Dallas), Shivvrat Arya (The University of Texas at Dallas ), Bharath Challa (The University of Texas at Dallas), Likhitha Pallapothula (University of Texas at Dallas ), AKSHAY VYAS (University of Texas at Dallas), Qifan Zhang (The University of Texas at Dallas), Jikai Wang (University of Texas at Dallas), Vasundhara Komaragiri (UT Dallas), Eric Ragan (University of Florida), Nicholas Ruozzi (UT Dallas), Yu Xiang (The University of Texas at Dallas), Vibhav Gogate (UT Dallas), Shin'ya Yamaguchi (NTT / Kyoto University), Daiki Chijiwa (NTT), Sekitoshi Kanai (NTT), Atsutoshi Kumagai (NTT Computer and Data Science Laboratories), Hisashi Kashima (Kyoto University), Jiaheng Wei (UCSC), Zhaowei Zhu (University of California, Tianyi Luo (Amazon), Ehsan Amid (Google Brain), Abhishek Kumar (Google Brain), Muhammed T Razzak (University of Oxford), Anthony

Ortiz (Microsoft), Caleb Robinson (Microsoft AI for Good Research Lab), Fangyi Chen (Carnegie Mellon University), Han Zhang (CMU), Hao Chen (Carnegie Mellon University), Kai Hu (Carnegie Mellon University), Jiachen Dou (Carnegie Mellon University), zaiwang li (pitt), Chenchen Zhu (Meta), Marios Savvides (Carnegie Mellon University), A. Feder Cooper (Cornell University), Wentao Guo (Cornell University), Duc Khiem Pham (Cornell University), Tiancheng Yuan (Cornell University), Charlie F Ruan (Cornell University), Yucheng Lu (Cornell University), Christopher De Sa (Cornell University), Rasool Fakoor (AWS), Zachary Lipton (Carnegie Mellon University), Pratik A Chaudhari (University of Pennsylvania), Alex J Smola (Amazon), Mark Vero (ETH Zurich), Mislav Balunovic (ETH Zurich), Martin Vechev (ETH Zurich), Jinyi Liu (Tianjin University), Yi Ma (Tianjin University), Jianye Hao (Tianjin University), Yujing Hu (NetEase Fuxi AI Lab), Yan Zheng (Tianjin University), Tangjie Lv (NetEase Fuxi AI Lab), Changjie Fan (NetEase Fuxi AI Lab), Gregory Yauney (Cornell University), Emily Reif (Google), David Mimno (Cornell University), Hailey Joren (UC San Diego), Chirag Nagpal (Carnegie Mellon University), Katherine Heller (Google), Berk Ustun (UCSD), Alex Oesterling (Harvard University), Jiaqi Ma (University of Illinois Urbana-Champaign), Flavio Calmon (Harvard University), Himabindu Lakkaraju (Harvard), Luísa B Shimabucoro (Universidade de São Paulo), Timothy Hospedales (Edinburgh University), Henry Gouk (University of Edinburgh), Pratyush Maini (IIT Delhi), Sachin Goyal (Carnegie Mellon University), Zico Kolter (Carnegie Mellon University), Aditi Raghunathan (Carnegie Mellon University), Aaditya Naik (University of Pennsylvania), Yinjun Wu (University of Pennsylvania), Mayur Naik (University of Pennsylvania), Eric Wong (University of Pennsylvania), Karthick Gunasekaran (Researcher), Sang Keun Choe (Carnegie Mellon University), Sanket Vaibhav Mehta (Carnegie Mellon University), Hwijeen Ahn (Carnegie Mellon University), Willie Neiswanger (Stanford University), Pengtao Xie (UC San Diego), Emma Strubell (Carnegie Mellon University), Eric Xing (MBZUAI, CMU, and Petuum Inc.), Guozheng Ma (Tsinghua University), Linrui Zhang (Tsinghua University), Haoyu Wang (Tsinghua University), Lu Li (Tsinghua University), Zilin Wang (Tsinghua University), Zhen Wang (The University of Sydney ), Li Shen (JD Explore Academy), Xueqian Wang (Tsinghua University), Dacheng Tao (The University of Sydney), Yi-Fan Zhang (NLPR, Xue Wang (Alibaba DAMO Academy), Weiqi Chen (Alibaba Group), Zhang Zhang (Institute of Automation, Rong Jin (Twitter), Tieniu Tan (NLPR, Jiachen T. Wang (Princeton University), Yuqing Zhu (UC Santa Barbara), Yu-Xiang Wang (UC Santa Barbara), Prateek Mittal (Princeton University), Ching-Yun Ko (MIT), Pin-Yu Chen (IBM Research), Payel Das (IBM Research), Yung-Sung Chuang (MIT), Luca Daniel (Massachusetts Institute of Technology), Young In Kim (Purdue University), Pratiksha Agrawal (Purdue University), Johannes Royset (Naval Postgraduate School), Rajiv Khanna (Purdue University), Megan Richards (Meta), Diane Bouchacourt (Meta), Mark Ibrahim (Meta), Polina Kirichenko (New York University), Chiyuan Zhang (MIT), Linus Ericsson (University of Edinburgh), Newsha Ardalani (Meta AI (FAIR)), Mostafa Elhoushi (Meta), Carole-Jean Wu (Meta AI), Jacob Buckman (Mila), Kshitij Gupta (Mila), Ethan Caballero (Mila), Rishabh Agarwal (Google Research, Brain Team), Marc G. Bellemare (Google Brain), Avni Kothari (UC San Diego), Lily Weng (UCSD), Bogdan Kulynych (EPFL), Yoav Wald (Johns Hopkins), Suchi Saria (Johns Hopkins University), Hanyang Jiang (Georgia Institute of Technology), Yao Xie (Georgia Tech), Ellen Zegura (Georgia Tech), Elizabeth Belding (University of California, Santa Barbara), Shaowu Yuchi

(Georgia Institute of Technology), Kaize Ding (Arizona State University), Yancheng Wang (Arizona State University), Huan Liu (Arizona State University), Jeyeon Eo (Soongsil University), Dongsu Lee (Soongsil University ), Minhae Kwon (Soongsil University), Thao T Nguyen (University of Washington), Samir Gadre (Columbia University), Gabriel Ilharco (University of Washington), Sewoong Oh (University of Washington), Kimia Hamidieh (University of Toronto, Vector Institute), Haoran Zhang (MIT), Thomas Hartvigsen (MIT), Marzyeh Ghassemi (University of Toronto, Amro Abbas (Meta), Surya Ganguli (Stanford University), Hidetomo Sakaino (Weathernews Inc.)

## Appendix B. Full list of accepted papers

The full list of accepted papers is available at `https://dmlr.ai/accepted/`.

## Appendix C. Links to recorded talks and slides from DMLR

In random order:

Masashi Sugiyama
*Coping with Wild Distribution Shifts: Continuous Shift, Joint Shift, and Beyond*
https://slideslive.com/39006435/coping-with-wild-distribution-shifts-continuous-shift-joint-shift-and-beyond?ref=folder-122509

Ce Zhang
*DMLR: Journal of Data-centric Machine Learning Research*
https://slideslive.com/39006439/dmlr-journal-of-datacentric-machine-learning-research?ref=folder-122509

Dina Machuve
*Data for Agriculture: Challenges and Opportunities in East Africa*
https://slideslive.com/39006438/data-for-agriculture-challenges-and-opportunities-in-east-africa?ref=folder-122509

Peter Mattson
*Data-centric Ecosystem: Croissant and Dataperf*
https://slideslive.com/39006431/datacentric-ecosystem-croissant-and-dataperf?ref=folder-122509

Olga Russakovsky and Vikram Ramaswamy
*Data-centric Machine Learning: Tackling social bias in computer vision datasets*
https://slideslive.com/39006434/datacentric-machine-learning-tackling-social-bias-in-computer-vision-datasets?ref=folder-122509

Andrew Ng
*Fast prompt-based ML development and data-centric AI*
https://slideslive.com/39006430/fast-promptbased-ml-development-and-datacentric-ai?ref=folder-122509

Ludwig Schmidt, Megan Ansdell, Nathan Lambert, Sang Michael Xie, Praveen Paritosh, Manil Maskey
*Panel Discussion*
https://slideslive.com/39006440/panel-discussion?ref=folder-122509

Mihaela van der Schaar
*Reality-Centric AI*
https://slideslive.com/39006433/realitycentric-ai?ref=folder-122509

Isabelle Guyon
*Towards Data-Centric AutoML*
https://slideslive.com/39006437/towards-datacentric-automl?ref=folder-122509

