# OpenReview forum: "DMLR: Data-centric Machine Learning Research - Past, Present and Future"
_DMLR — Accepted by DMLR_

### Review · Reviewer_6sgG · 2024-02-02

**Recommendation:** 3
**Confidence:** 1

**Summary Of Contributions:**

This paper is a comprehensive review of the evolution and current state of data-centric approaches in machine learning research. The paper begins by addressing the historical ambivalence in machine learning towards data, highlighting the shifts between algorithm-focused and data-focused approaches. It then discusses significant moments and contributions in the field, emphasizing the importance of data in developing effective machine-learning models. Overall, the paper serves as a call to action for the machine learning community to focus more on data-centric approaches, recognizing the crucial role of high-quality datasets in advancing the field.

**Strengths:**

**Relation to Prior Work**: It effectively contextualizes its findings within the broader scope of past research, showing a clear understanding of the field's evolution.

**Relevance to Research Community**: The focus on data-centric approaches is highly relevant to current trends and challenges in machine learning research.

**Clarity of Presentation**: The paper is well-structured and clearly written, making complex ideas accessible.

**Audience:**

Yes

**Claims And Evidence:**

Yes

**Datasets And Benchmarks:**

N/A

**Extended Submissions:**

N/A

**Limitations:**

See **Strengths And Weaknesses**

**Requested Changes:**

**Interaction with Data**: While the paper focuses on data, it could further explore how different types of data interact within the realm of machine learning, offering a more dynamic perspective.

**Concrete Examples Across Domains**: The inclusion of concrete examples in various domains like computer vision, remote sensing, medical, and agriculture would enrich the paper by demonstrating the practical applications and implications of the discussed concepts in diverse fields.

**Strengths And Weaknesses:**

**S1**: Comprehensive Coverage: The paper thoroughly explores the historical and current importance of data in machine learning.

**S2**: Emphasis on Community Engagement: Advocating for a community-driven approach to improving datasets is a significant and progressive step.



**W1**:Narrow Focus: The paper primarily concentrates on data-centric methods in machine learning, which might not fully address other essential aspects like the interaction between data and algorithm development.

**W2**: Need for a Balanced Perspective: A more comprehensive discussion on the limitations and challenges associated with data-centric approaches in machine learning would enhance the paper's balance and depth.

---

### Review · Reviewer_tK1e · 2024-02-18

**Recommendation:** 4
**Confidence:** 1

**Summary Of Contributions:**

This paper is an editorial that summarizes important knowledge about the past, present and future of the DMLR community. This is also a good starting point for potential researchers to get to know data-centric researches and get involved.

**Strengths:**

[+] This paper is well-written and easy to follow.
[+] This editorial summarizes the important milestones for the data-centric researches. It is beneficial for potential researchers in data-centric areas.
[+] The figures are well-designed.

**Audience:**

Yes

**Broader Impact Concerns:**

no concerns

**Claims And Evidence:**

yes

**Datasets And Benchmarks:**

not dataset or benchmark

**Extended Submissions:**

no

**Limitations:**

[-] Overall, it is hard to identify any weakness of this paper. However, one comment is that, it might be better to highlight the vision or definition of the next generation of machine learning datasets, even though it has been introduced in page 3. Or some examples will be even better for readers to understand this idea.

**Requested Changes:**

no

**Strengths And Weaknesses:**

[+] This paper is well-written and easy to follow.
[+] This editorial summarizes the important milestones for the data-centric researches. It is beneficial for potential researchers in data-centric areas.
[+] The figures are well-designed.

---

### Review · Reviewer_tD36 · 2024-03-01

**Recommendation:** 3
**Confidence:** 3

**Summary Of Contributions:**

This article delves into the advancements in data-centric machine learning, covering its historical development, current state, and future prospects. The authors connect benchmarks, datasets, relevant infrastructures, and advancements in modern models to create an ecosystem for research in data-centric machine learning. Additionally, the paper aims to promote greater engagement from DMLR communities in fostering an open, collaborative, and inclusive exchange of diverse perspectives.

**Strengths:**

The paper provides a fundamental contribution to the community through clear and impactful writing and research on outlining changes. Its influence can extend to the broader research community, fostering increased engagement. The strengths of the paper lie in its broader impact and fundamental research.

**Audience:**

Yes

**Broader Impact Concerns:**

The paper has outlined the broader impact on the DMLR community. Concerns regarding data ownership and trust should be taken into consideration.

**Claims And Evidence:**

The authors of the paper acknowledge the possibility of approximation errors stemming from different perspectives and are open to making changes and updates. Overall, I find their claims to be correct and accurate.

**Datasets And Benchmarks:**

The paper does not focus on datasets and benchmarks, but rather provides an overview of the changes in DMLR development. Additionally, the authors include rich web link sources to connect to numerous open and well-known benchmarks and datasets collections.

It's not a datasets and benchmarks related papers. It overviews the changes in DMLR development. But in the paper, the autors provide rich web link sources to connect to many open- and well-known- benchmarks or datasets collection.

**Extended Submissions:**

Based on the information I have, I don't believe it is an extended version of a previously published work.

**Limitations:**

This paper serves as a review of DMLR development and proposes ideas for the future ecosystem. I did not identify any weaknesses and limitations in their writing.

**Requested Changes:**

I am interested in gaining insights into data trust, including aspects such as data provenance and ownership. It would be beneficial to provide more commentary on this topic and establish connections to the proposed ecosystem for future public datasets.

**Strengths And Weaknesses:**

Strengths:
The paper significantly contributes to the community with its clear and impactful writing and research on outlining changes. It has the potential to influence the broader research community, encouraging increased engagement. The strengths of the paper are evident in its broader impact and foundational research.

Weaknesses:
This paper offers a comprehensive review of the evolution of DMLR and introduces a new ecosystem to the community. No shortcomings were found in the writing.